# Cryo-electron tomography of Birbeck granules reveals the molecular mechanism of langerin lattice formation

Toshiyuki Oda[1]*, Haruaki Yanagisawa[2], Hideyuki Shinmori[3], Youichi Ogawa[4], Tatsuyoshi Kawamura[4]

[1]Department of Anatomy and Structural Biology, Graduate School of Medicine, University of Yamanashi, Chuo, Japan; [2]Department of Cell Biology and Anatomy, Graduate School of Medicine, University of Tokyo, Tokyo, Japan; [3]Faculty of Life and Environmental Science, University of Yamanashi, Kofu, Japan; [4]Department of Dermatology, University of Yamanashi, Chuo, Japan

**Abstract** Langerhans cells are specialized antigen-presenting cells localized within the epidermis and mucosal epithelium. Upon contact with Langerhans cells, pathogens are captured by the C-type lectin langerin and internalized into a structurally unique vesicle known as a Birbeck granule. Although the immunological role of Langerhans cells and Birbeck granules have been extensively studied, the mechanism by which the characteristic zippered membrane structure of Birbeck granules is formed remains elusive. In this study, we observed isolated Birbeck granules using cryo-electron tomography and reconstructed the 3D structure of the repeating unit of the honeycomb lattice of langerin at 6.4 Å resolution. We found that the interaction between the two langerin trimers was mediated by docking the flexible loop at residues 258–263 into the secondary carbohydrate-binding cleft. Mutations within the loop inhibited Birbeck granule formation and the internalization of HIV pseudovirus. These findings suggest a molecular mechanism for membrane zippering during Birbeck granule biogenesis and provide insight into the role of langerin in the defense against viral infection.

*For correspondence: toda@yamanashi.ac.jp

Competing interest: The authors declare that no competing interests exist.

## Editor's evaluation

Langerhans cells, found in the epidermis and mucosal epithelium, capture pathogens via a C-type lectin called langerin. The capture leads to the formation of a membrane vesicle called a Birbeck granule. Here the authors use cryo-electron tomography to show, for the first time, how langerin forms a zippered lattice. They use their structure to design mutations to disrupt lattice formation and show that this interferes with HIV internalization.

## Introduction

The epidermis and mucosa provide the first line of defense against pathogens. In these tissues, specialized antigen-presenting cells called Langerhans cells capture pathogens by recognizing their surface mannose-containing oligosaccharides via the C-type lectin langerin (*Valladeau et al., 2003*; *Doebel et al., 2017*). Pathogens internalized via langerin are enclosed in a specialized organelle called a Birbeck granule, which has a central striated lamella between closely apposed membranes (*Birbeck et al., 1961*; *Valladeau et al., 2000*; *McDermott et al., 2004*; *Valladeau et al., 1999*). Once pathogens are captured, Langerhans cells undergo maturation and migrate to peripheral lymph nodes, where they present processed antigens to T cells (*Merad et al., 2008*; *de Jong et al., 2010*; *Rhodes et al.,*

*2021*). Langerin is a type II transmembrane protein that is composed of a cytoplasmic tail, coiled-coil neck region, and carbohydrate recognition domain (CRD) (*Stambach and Taylor, 2003*; *Valladeau et al., 1999*; *Valladeau et al., 2000*; *Figure 1A*). Through the coiled-coil neck region, langerin forms a trimer, which is functionally important for oligosaccharide binding (*Stambach and Taylor, 2003*; *Weis and Drickamer, 1996*; *Weis and Drickamer, 1994*). The calcium-dependent carbohydrate-binding capacity of langerin CRD is essential for the formation of Birbeck granules because the removal of calcium ions causes them to unzip and disintegrate (*Stambach and Taylor, 2003*; *Bartosik, 1992*). It has been proposed that the 'face-to-face' interaction between the langerin trimers via CRDs confers the characteristic zippered membranes with a central striation of Birbeck granules (*Thépaut et al., 2009*' *Figure 1B and C*). However, the precise molecular mechanism of Birbeck granule formation is unknown because of the lack of a high-resolution structure.

Langerin's high affinity for mannose and oligo-mannose allows Langerhans cells to capture a wide range of pathogens, including *Saccharomyces cerevisiae, Candida albicans, Mycobacterium leprae,* herpes simplex virus, and human immunodeficiency virus-1 (HIV-1) (*Tateno et al., 2010*; *Hunger et al., 2004*; *de Jong and Geijtenbeek, 2010*; *de Witte et al., 2007*; *Turville et al., 2002*). Although Langerhans cells in the mucosa are the initial target of HIV-1 in sexually transmitted infections, the efficiency of Langerhans cell infection is low, requiring high viral loads of HIV-1 to infect cells (*Kawamura et al., 2001*; *Kawamura et al., 2000*; *Ogawa et al., 2013*). The resistance of Langerhans cells to HIV-1 infection is mediated by the efficient internalization of HIV-1 into Birbeck granules and subsequent autophagic degradation (*de Witte et al., 2007*; *Ribeiro et al., 2016*). Therefore, elucidating the mechanism by which HIV-1 is internalized into Birbeck granules is essential to understanding the role of Langerhans cells as a barrier to HIV-1 infection.

In the present study, we reconstructed the 3D structure of isolated Birbeck granules using cryo-electron tomography and identified a flexible loop region that is crucial for Birbeck granule formation and virus internalization.

## Results

### Cryo-electron microscopy of isolated Birbeck granules

Exogenous expression of langerin in mammalian cell lines readily reproduces the formation of Birbeck granules (*McDermott et al., 2004*). We found that the addition of yeast mannan to langerin-expressing cells enhances the formation of large Birbeck granules (*Figure 1C*). To purify these Birbeck granules, we expressed SNAP-tagged human langerin in 293T cells, biotinylated the tag by incubation with benzylguanine-biotin, and immunoprecipitated the Birbeck granules using streptavidin agarose (*Figure 1D* and *Figure 1—figure supplement 1A*). Under cryo-electron microscopy, the isolated Birbeck granules were observed to be thin lamellar vesicles of assorted sizes, similar to those reported previously (*Figure 1E and F*, and *Figure 1—figure supplement 1B*; *Sagebiel and Reed, 1968*; *Lenormand et al., 2013*). Unlike regular 2D crystals of proteins, Birbeck granules appear wavy, suggesting structural heterogeneity. The 2D classification of the segmented images extracted from the micrographs showed nearly three-fold symmetrical arrangements of pores with a diameter of approximately 10 nm; however, no clear crystalline lattice was observed (*Figure 1G*). Although most Birbeck granules were oriented in approximately the same direction with respect to the beam axis, some granules were twisted, providing different views that were sufficient for 3D reconstruction (*Figure 1—figure supplement 1C*).

### Subtomogram averaging of the repeating unit of Birbeck granules

We conducted cryo-electron tomography of the isolated Birbeck granules and reconstructed the repeating units by averaging the subtomograms (*Figure 2* and *Figure 2—figure supplement 1A*). The initial density map (*Figure 2A*) shows a langerin trimer (blue) with three inverted trimers (red) attached to the CRDs. However, because of the flexibility between each langerin trimer, the resolution was limited to ~24 Å. To improve the resolution, we generated a mask enclosing the central langerin trimer and three attached CRDs and performed focused refinement and 3D classification (*Figure 2B*). The resolution was improved to 6.4 Å (*Figure 2—figure supplement 1B*), which clearly visualized the alpha helices and enabled us to generate a model based on the previously reported crystal structure of the langerin trimer (*Feinberg et al., 2010*). We then performed a two-body refinement to visualize

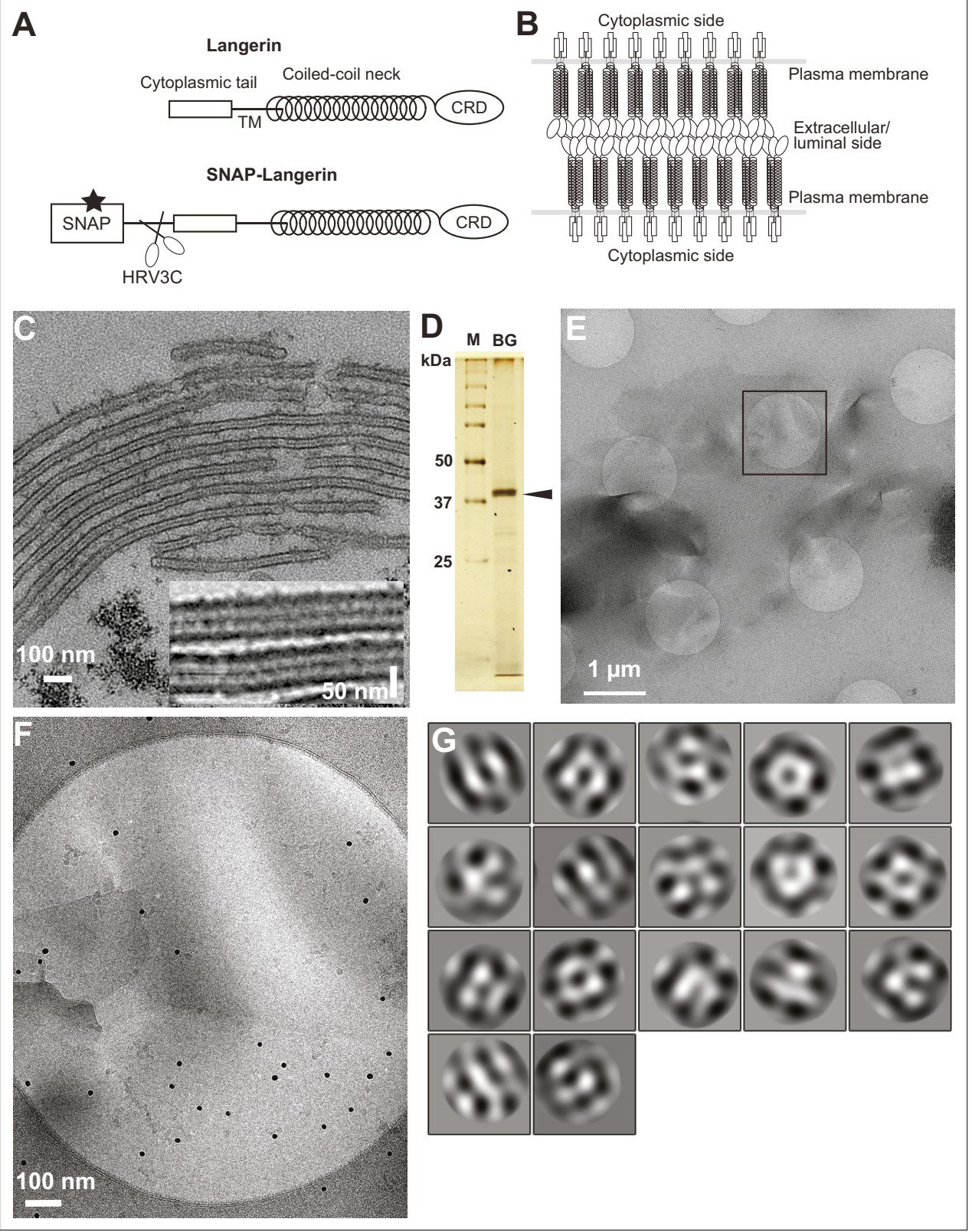

**Figure 1.** Isolation of Birbeck granules. (**A**) Domain organization of langerin. Human langerin is a 328-amino acids protein composed of an N-terminal cytoplasmic tail, a transmembrane domain (TM), a coiled-coil neck, and a C-terminal carbohydrate-recognition domain (CRD). We fused a SNAP-tag at the N-terminus of langerin and inserted an HRV3C protease cleavage site between the tag and the langerin sequences. For streptavidin-mediated precipitation of Birbeck granules, the SNAP-tag was biotinylated (star). (**B**) Model of langerin oligomerization within Birbeck granules. Langerin trimers

*Figure 1 continued on next page*

*Figure 1 continued*

bind to each other face-to-face via the CRDs, bringing the two layers of the plasma membrane closer together. (**C**) Birbeck granules formed in 293T cells overexpressing langerin. Addition of yeast mannan induced the formation of Birbeck granules a few micrometers long. Inset shows a magnified view of Birbeck granules. (**D**) SDS-PAGE of isolated Birbeck granules. Purified langerin (arrowhead) was released from streptavidin-agarose by HRV3C digestion. M: molecular weight marker; and BG: isolated Birbeck granules. (**E**) Cryo-electron microscopy of isolated Birbeck granules. Black square indicates the position of the close-up view shown in F. (**F**) Wavy lamellar structure of the Birbeck granule. Black dots were gold nanoparticles used as fiducial markers. (**G**) Class averages of the projection images of Birbeck granules. The image dimension is 34 nm². Although 2D classification did not converge well due to the continuity of the structure, some classes showed a porous structure with a honeycomb-like lattice.

The online version of this article includes the following source data and figure supplement(s) for figure 1:

**Source data 1.** Original gel image of *Figure 1D*.

**Source data 2.** Annotated gel image of *Figure 1D*.

**Figure supplement 1.** Isolation of Birbeck granules.

the spatial relationship between the two opposing langerin trimers (*Nakane et al., 2018*). Based on this refinement (*Figure 2C*), we built a honeycomb lattice model (*Figure 2D* and *Figure 2—video 1*), which agrees with the striation of the Birbeck granules observed by ultra-thin section electron micros- copy and the ~10 nm pores visualized in the 2D classification images (*Figure 1C and G*; *Birbeck et al., 1961*). However, in reality, the honeycomb lattice is non-uniform because of the structural heterogeneity.

## Interaction between the two langerin trimers via flexible loops

An in-depth analysis of the interaction region between the two CRDs (*Figure 3*) revealed that the interaction is mediated by a loop at residues 258–263, which is highly flexible in the crystal struc- ture (*Feinberg et al., 2010*). The interaction between the two CRDs appears to be maintained by docking the loops of two langerin molecules into each other's lateral cleft, which constitutes the secondary carbohydrate-binding site (*Figure 3A*, pink circles; *Chatwell et al., 2008*). The inter- action between the two CRDs is flexible because the first eigenvector of the multi-body anal- ysis represents an approximately 30° tilting motion with the loop as the fulcrum (*Figure 3B*, and *Figure 3—video 1*). Interestingly, loop 258–263 of the inverted molecule appears to participate in the secondary carbohydrate-binding site of the upright molecule (*Figure 3A*). A previous study found that the secondary calcium-independent binding site has a lower affinity for carbohydrate ligands than the primary calcium-dependent binding site (*Chatwell et al., 2008*). The docked loop 258–263 may provide additional interactions with the hydroxyl group of mannose via Glu261 and Asp263, thus increasing the affinity of the secondary binding site when two langerin trimers form a complex. Furthermore, the Trp264Arg mutation is associated with a lack of Birbeck granules (*Verdijk et al., 2005*; *Figdor et al., 2002*). Trp264 may also be involved in docking the loop into the cleft; however, it is more likely that disruption of the hydrophobic core of the CRD is responsible for this defect in Birbeck granule formation (*Chatwell et al., 2008*).

## Loop 258-263 is essential for Birbeck granule formation

To investigate the importance of loop 258–263 in the biogenesis of Birbeck granules, we mutated three residues in the loop: Met260Ala, Glu261Arg, and Asp263Lys, and generated three mutated langerin constructs, each carrying one (MRGD), two (MRGK), or three mutations (ARGK). We expressed these three langerin mutants in 293T cells and induced Birbeck granule formation by adding yeast mannan (*Figure 4A*). Under ultra-thin section microscopy, wild-type langerin produced Birbeck granules a few micrometers in length in the presence of mannan (*Figure 4A*, MEGD WT). As the number of muta- tions increased, the length of Birbeck granules decreased. With two point mutations (MRGK), some cross-sections became spherical rather than rod-shaped, suggesting that the Birbeck granules failed to extend. With three mutations (ARGK), clusters of Birbeck granules were not observed, with isolated individual granules observed instead. The length of individual Birbeck granules and the total length of granules per area were measured to confirm the above observations (*Figure 4B and C*). In contrast, the number of the Birbeck granules showed less significant differences because numerous short frag- ments were present in the mutant specimen (*Figure 4D*). These results demonstrate the importance of loop 258–263 in the biogenesis of Birbeck granules.

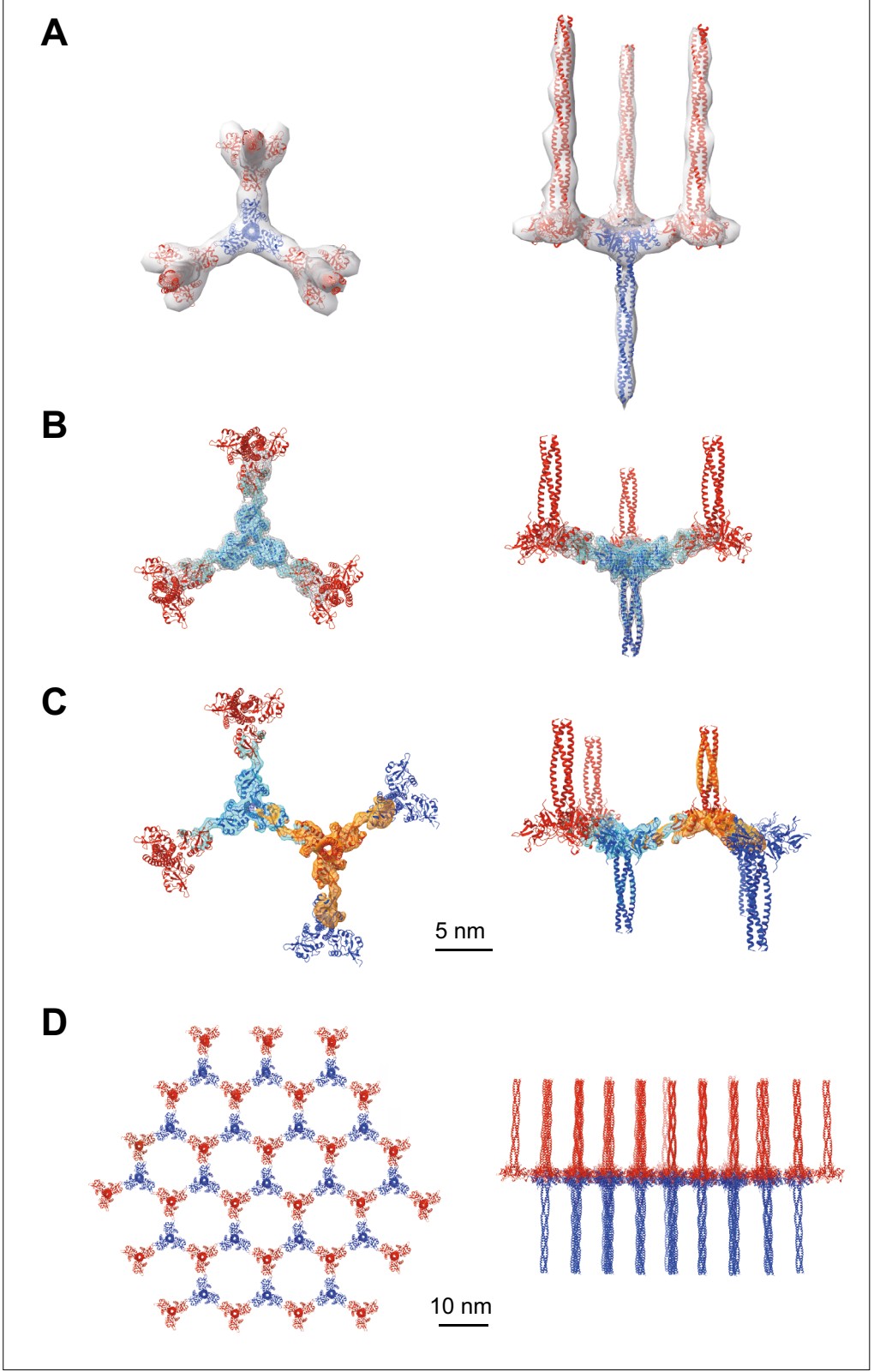

**Figure 2.** Averaged subtomograms of Birbeck granules. (**A**) Result of the initial 3D refinement. The central langerin trimer (blue) binds to the three inverted langerin trimers (red). (**B**) Result of the focused refinement. Improved resolution (6.4 Å) allowed secondary structure modeling. The gray mesh and the cyan surface indicate low and high threshold isosurfaces, respectively. (**C**) Result of the two-body refinement. The second body (orange) was refined

*Figure 2 continued on next page*

*Figure 2 continued*

with respect to the first body (cyan). (**D**) Honeycomb model of the langerin lattice within Birbeck granules. This honeycomb lattice is an ideal model, with the assumption of structural uniformity. (**A–D**) left, top views; right, side views.

The online version of this article includes the following video and figure supplement(s) for figure 2:

**Figure supplement 1.** Cryo-electron tomography of Birbeck granules.

**Figure 2—video 1.** Movie representation of the langerin lattice.

https://elifesciences.org/articles/79990/figures#fig2video1

## Loop 258-263 is crucial for the internalization of HIV pseudovirus

To investigate the impact of the mutations in loop 258–263 on virus internalization by Birbeck granules, we added lentiviruses pseudo-typed with the HIV-1 env protein to langerin-expressing 293T cells. After 30 min of incubation, we retrieved viruses attached to the surface langerin by washing with EDTA buffer, removed non-specifically attached viruses by trypsinization, and isolated the Birbeck granules using streptavidin-agarose (*Figure 5*). A langerin mutant that lost its calcium-binding capacity (*Stambach and Taylor, 2003*) was used as the negative control (*Figure 5*, lectin (-)). The HIV pseudoviruses attached to both wild-type and ARGK mutant langerin and the attachment was abolished by the addition of mannan (*Figure 5A*). This indicated that the mutation of the loop did not disrupt the binding of the glycosylated envelope proteins to the primary carbohydrate-binding sites. In contrast, the internalization of the pseudovirus was significantly inhibited by the ARGK mutation (*Figure 5B and C*). These results suggest that disruption of Birbeck granule formation by the mutation of loop 258–263 inhibited the internalization of HIV pseudovirus.

## Discussion

### Virus internalization mechanism of Birbeck granules

In the present study, we reconstructed the 3D structure of isolated Birbeck granules and found that docking of the flexible loop 258–263 into the secondary carbohydrate-binding cleft is essential for assembly of the langerin lattice. In agreement with this observation, we showed that mutations introduced into the loop inhibited Birbeck granule formation and internalization of HIV pseudovirus. Based on the structure of the glycosylated HIV env protein (*Stewart-Jones et al., 2016*; *Pan et al., 2020*), we propose a model of HIV internalization by langerin (*Figure 6A and B*). A previous study proposed that N-linked high-mannose oligosaccharides bind to langerin at both primary and secondary carbohydrate-binding sites (*Chatwell et al., 2008*). Our model requires bending of the coiled-coil necks of the two langerin trimers to form a complex with HIV-1 env via the high-mannose oligosaccharide (*Figure 6B*). Although the triple coiled-coil structure of the neck domain does not seem to bend strongly, the coiled-coil prediction profile suggests that the N-terminal part of the neck does not necessarily form a rigid triple coiled coil (*Lupas et al., 1991*; *Figure 1—figure supplement 1D*). Thus, we speculate that the neck region of the langerin trimer is flexible and swings on the cell surface, allowing loop 258–263 of one trimer to dock into the cleft of an adjacent trimer. Once the triad structure, consisting of two langerin trimers and one env protein, is formed, the elasticity of the neck region can deform the cell membrane and initiate invagination. The formation of clathrin pits may also be involved in membrane invagination, because the cytoplasmic ends of Birbeck granules are coated with clathrin (*Mc Dermott et al., 2002*). Additionally, it is possible that other intracellular mechanisms such as the actin cytoskeleton, are involved in this membrane deformation process because annexin A2 is required for the proper formation of Birbeck granules (*Thornton et al., 2020*).

### Significance of langerin-specific secondary carbohydrate-binding cleft

It is noteworthy that the docking of loop 258–263 into the secondary calcium-independent carbohydrate-binding cleft, which is not found in other C-type lectins (*Chatwell et al., 2008*), mediates the assembly of the langerin lattice. In DC-SIGN, which is another C-type lectin expressed in dendritic cells, and is closely related to langerin, the region corresponding to the secondary carbohydrate-binding site of langerin forms a shallow pocket, that is occupied by two calcium ions (*Figure 7A*; *Guo et al., 2004*; *Geijtenbeek et al., 2000a*; *Geijtenbeek et al., 2000b*). This structural difference between

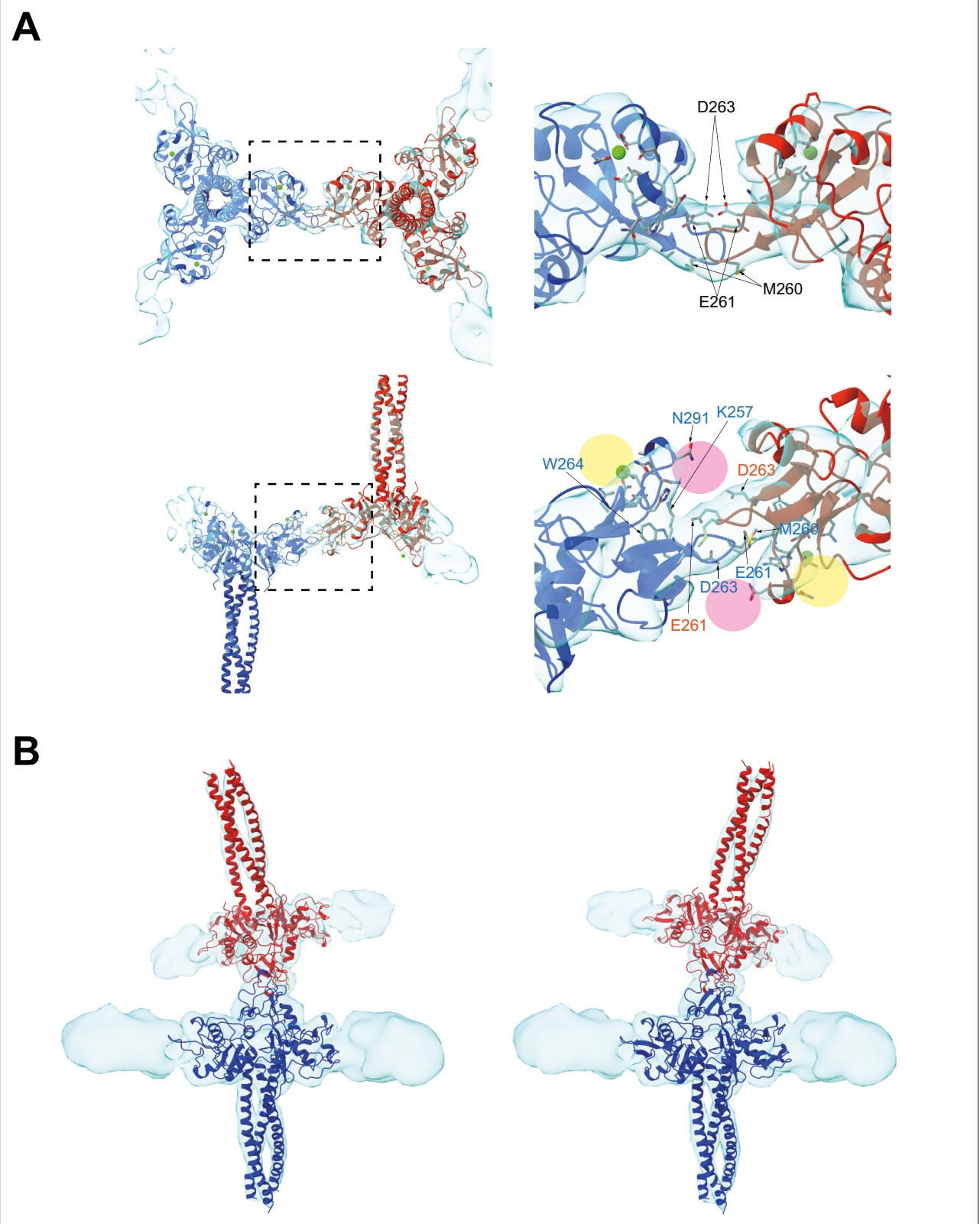

**Figure 3.** Binding interface between two langerin trimers. (**A**) Two langerin trimers interact via the loops at residues 258–263. Amino acid codes and residue numbers are indicated. Yellow and pink circles indicate the primary and secondary carbohydrate-binding sites. The orientations of the side chains are speculative because the resolution of these maps (6.4 Å) was insufficient for precise modeling. Calcium ions (green) were placed based on the crystal structure (PDB ID: 3KQG). Top and side views are displayed in the top and bottom rows, respectively. Broken squares indicate the positions of the

*Figure 3 continued on next page*

*Figure 3 continued*

magnified views on the right. (**B**) Flexibility between the two trimers. The first eigenvector of the structural variation of the inverted trimer (red) relative to the upright trimer (blue) shows an approximately 30° rotation about the loop 258–263 (see *Figure 3—video 1*).

The online version of this article includes the following video for figure 3:

**Figure 3—video 1.** Morphing movie showing the first eigenvector of structural variation calculated by multibody analysis.

https://elifesciences.org/articles/79990/figures#fig3video1

langerin and DC-SIGN may account for the inability of DC-SIGN to form Birbeck granules despite its high sequence similarity to langerin (*Cambi et al., 2009*). Another characteristic of the secondary carbohydrate-binding site of langerin is its weak affinity for mannose and maltose (*Chatwell et al., 2008*). This may reflect that the sugar-binding surface of langerin CRD is less negatively charged than that of DC-SIGN CRD (*Figure 7A*; *Chatwell et al., 2008*). Moreover, the interaction between the secondary binding site and mannose/maltose is mainly mediated by backbone amide and carbonyl groups (*Chatwell et al., 2008*). It is possible that the side chains of Glu261 and Asp263 of loop 258–263 docked into the cleft of an adjacent langerin CRD provide the negative charge required for stabilizing oligo-mannose bound to the secondary binding site (*Figure 7A and B*). In agreement with this hypothesis, in the present study the Asp263Lys mutation caused a significant defect in Birbeck granule formation. The secondary carbohydrate-binding site may only be fully functional when two langerin trimers are bound to each other (*Figures 6 and 7B*). The effect of the Met260Ala mutation on Birbeck granule formation was unexpected, because the mutation was supposed to be silent in terms of side chain charge. The observed structural variation of the langerin trimer complex suggests that the side chain of Met260 acts as a hook to stabilize the interface between the two intersecting loops. Note that we could not visualize a oligo-mannose bound to the secondary binding site in our maps. A higher resolution map is required to prove our carbohydrate-recognition model.

Our model of Birbeck granule formation suggests that oligosaccharide ligands are captured by the lateral surface of langerin (*Figure 6*). In contrast, DC-SIGN binds to dengue virus on its upper surface (*Pokidysheva et al., 2006*). The difference in the ligand-binding manner between langerin and DC-SIGN may contribute to their different carbohydrate recognition preferences (*Valverde et al., 2020*; *Takahara et al., 2004*). Further studies are needed to investigate the physiological significance of the unique carbohydrate-binding mechanism of langerin and the honeycomb lattice structure of Birbeck granules.

## Materials and methods
### Cell lines
We purchased Lenti-X 293T (product code: 632180) and HEK293 cell lines (CRL-1573) from TakaraBio, Tokyo, Japan and ATCC, respectively. Cell lines were authenticated by analysis of short tandem repeat profiling (BEX, Tokyo, Japan) and confirmed as negative for mycoplasma contamination using Myco-plasma Detection Kit (InvivoGen, San Diego, CA).

### Isolation of Birbeck granules
The cDNA of human langerin was obtained from the Mammalian Gene Collection library (Dnaform Inc, Yokohama, Japan) and was subcloned into pcDNA3.1-IRES-GFP (*Schaefer et al., 2008*). A codon-optimized sequence coding SNAP-3 × HA-HRV3C cleavage site (obtained from ThermoFisher Scientific, Waltham, MA) was inserted immediately before the start codon of the langerin sequence (*Cole, 2013*). $O^6$-benzylguanine-biotin was obtained by reacting 16 mM $O^6$-benzylguanine-amine (Accela ChemBio, Shanghai, China) with 50 mM biotin-$AC_5$-Osu (Dojindo, Kumamoto, Japan) and 30 mM triethylamine in dimethylformamide for 24 hr at 30 °C. Lenti-X 293T cells (TakaraBio) were trans-fected with the langerin-expression plasmid by electroporation using ECM630 electroporator (BTX, Holliston, MA) and seeded onto the polyethylenimine-coated dish (PEI, obtained from Fujifilm Wako Chemicals, Osaka, Japan). After 48 hr culturing in DMEM + 10% fetal bovine serum (FBS), the culture medium was replaced with serum-free DMEM containing 10 μM $O^6$-benzylguanine-biotin and 10 μg/ml yeast mannan (Nacalai tesque, Kyoto, Japan) and the cells were incubated for 30 min at 37 °C. The cells were then collected by scraping and washed five times with isotonic HMC buffer (Hepes 30 mM,

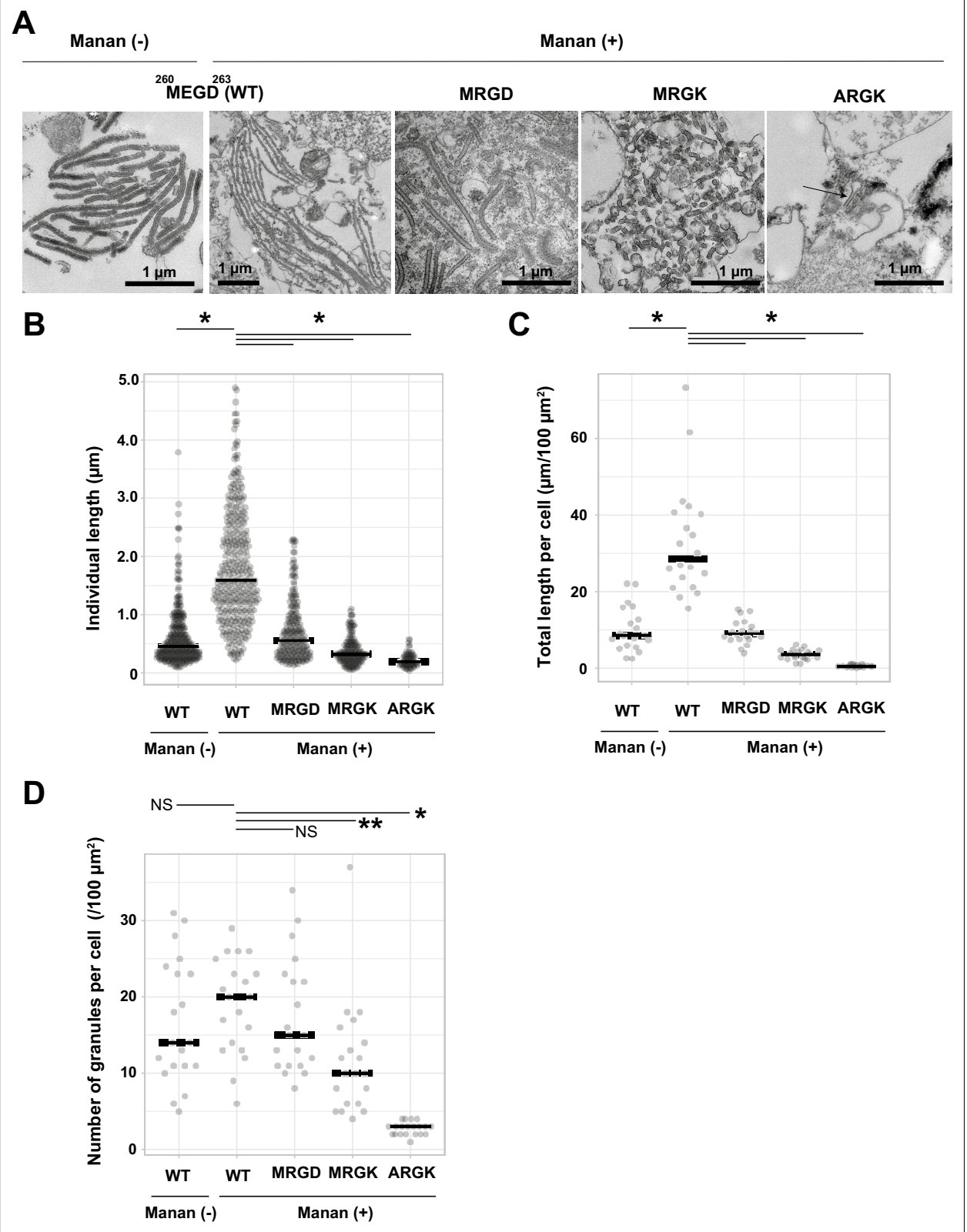

**Figure 4.** Mutations to the loop 258–263 disrupted Birbeck granule formation. (**A**) Ultra-thin electron microscopy of Birbeck granules with and without yeast mannan in langerin-overexpressing 293T cells. Residues at 260–263 (Met-Glu-Gly-Asp, MEGD) were mutated to Met-Arg-Gly-Asp (MRGD), Met-Arg-Gly-Lys (MRGK), or Ala-Arg-Gly-Lys (ARGK). The Mannan (+) WT image has a lower magnification than the others in order to show the entire Birbeck granules. (**B–D**) Quantification of Birbeck granule formation. Horizontal lines indicate the median values. Single asterisks and the double

*Figure 4 continued on next page*

*Figure 4 continued*

asterisk indicate statistically significant differences with p<0.01 and p=0.0101, respectively. NS indicates no statistically significant differences. p values were calculated using Bonferroni-corrected Student's *t*-tests. (**B**) Lengths of individual Birbeck granules were measured. N=337 (WT mannan-), 386 (WT mannan+), 348 (MRGD), 233 (MRGK), and 70 (ARGK). $p=1.4 \times 10^{-68}$ (WT mannan (-)), $2.6 \times 10^{-46}$ (MRGD), $6.6 \times 10^{-62}$ (MRGK), and $1.1 \times 10^{-35}$ (ARGK). (**C**) Sum of the length of Birbeck granules per 100 $\mu m^2$ cell area. Cross-sections of cells with nuclei of >5 $\mu m$ in diameter were selected and the total lengths of the Birbeck granules were measured within the cells. N=20 for all the samples. $p=2.9 \times 10^{-7}$ (WT mannan (-)), $5.0 \times 10^{-8}$ (MRGD), $1.5 \times 10^{-10}$ (MRGK), and $1.0 \times 10^{-11}$ (ARGK). (**D**) Number of the Birbeck granules per 100 $\mu m^2$ cell area. N=20 for all the samples. p=0.85 (WT mannan (-)), 1.2 (MRGD), 0.005 (MRGK), and $2.2 \times 10^{-13}$ (ARGK).

pH 7.2, 150 mM NaCl, 2 mM $MgCl_2$, 2 mM $CaCl_2$, protease inhibitor cocktail (Nacalai tesque)). The washed cells were homogenized with a Dounce glass homogenizer and were further disrupted by sonication for 8 s using a Q125 sonicator (Qsonica, Newtown, CT). The cell homogenate was loaded on top of a discontinuous Opti-prep density gradient (10-20%–40% layers, ThermoFisher Scientific) and ultracentrifuged for 1 hr at 100,000×g. The fraction at the interface of 20/40% layers was collected and diluted 10-fold with HMC buffer and was again disrupted by sonication for 2 min (120 cycles of 1 s pulse and 3 s wait) at 4 °C. Large debris was removed by filtration through a 20 µm-pore polyethylene mesh (GL Science, Tokyo, Japan) and the cell lysate was incubated with a 30 µl slurry of streptavidin agarose (Solulink, San Diego, CA) overnight at 4°C. The streptavidin agarose beads were washed five times with HMC buffer and the bound Birbeck granules were eluted by overnight digestion using home-made *E. coli* overexpressed His ×6-HRV3C protease in HMC buffer at 4 °C. After digestion and elution, HRV3C protease was removed by incubating the elute with Ni-NTA resin (Qiagen, Düsseldorf, Germany). About 50 µl of elute containing released Birbeck granules (approximately 0.05 mg/ml) were obtained from twelve 100 mm dishes of cell culture.

## Cryo-electron microscopy of isolated Birbeck granules

Five µl of the purified Birbeck granules was applied onto freshly glow-discharged holey carbon grids, Quantifoil R1.2/1.3 Cu/Rh 200 mesh (Quantifoil Micro Tools GmbH, Großlöbichau, Germany), blotted from the backside to concentrate the granules by filtration through the holes, and 4 µl of HMC buffer containing cytochrome *c*-stabilized 15 nm colloidal gold (BBI Solutions) was applied for attachment of fiducial markers (*Oda et al., 2015*). Grids were blotted from the backside for 10 s at 4 °C under 99% humidity and plunge frozen in liquid ethane using Vitrobot Mark IV (Thermo Fisher Scientific). We removed one of the blotting arms for one-side blotting because Birbeck granules would be absorbed onto the filter paper. Images were recorded using a Titan Krios G3i microscope at the University of Tokyo (Thermo Fisher Scientific) at 300 keV, a Gatan Quantum-LS Energy Filter (Gatan, Pleasanton, CA) with a slit width of 20 eV, and a Gatan K3 Summit direct electron detector in the electron counting mode. The nominal magnification was set to 33,000×with a physical pixel size of 2.67 Å/pixel. Movies were acquired using the SerialEM software (*Mastronarde, 2005*), and the target defocus was set to 3–6 µm. The angular range of the tilt series was from –60° to 60° with 3.0° increments. Each movie was recorded for 0.74 sec with a total dose of 1.24 electrons/$Å^2$ and subdivided into 20 frames. The total dose for one tilt series acquisition is thus 50 electrons/$Å^2$.

## Data processing

Movies were subjected to beam-induced motion correction using MotionCor2 (*Zheng et al., 2017*), and tilt series images were aligned, CTF corrected, and back-projected to reconstruct 3D tomograms using the IMOD software package (*Kremer et al., 1996*). For the subsequent analysis using Relion-4.0 (*Kimanius et al., 2021*), the tomogram positioning step was skipped to keep the angle offset and the Z-shift zero. Tomograms were 4×binned to reduce loads of the calculation. The areas of the Birbeck granules were manually selected using the 3dmod model tool (*Kremer et al., 1996*), and the center coordinates for the subtomograms were determined by placing points at 8-pixel (~8.5 nm) intervals within the selected area using a custom-made Ruby script (*Supplementary file 1*, *Metlagel et al., 2007*). For twisting Birbeck granules (*Figure 1—figure supplement 1C*), center points were picked manually by selecting the mid-portion of the lamella. Volumes with 32×32 × 32 pixel-dimensions were extracted from 8×binned tomograms and were averaged using the PEET software suite (*Nicastro et al., 2006*). A randomly selected subtomogram was used for the initial reference. Alignments were repeated five times for 8×binned and twice for 4×binned tomograms with 64×64 × 64 pixel-dimensions.

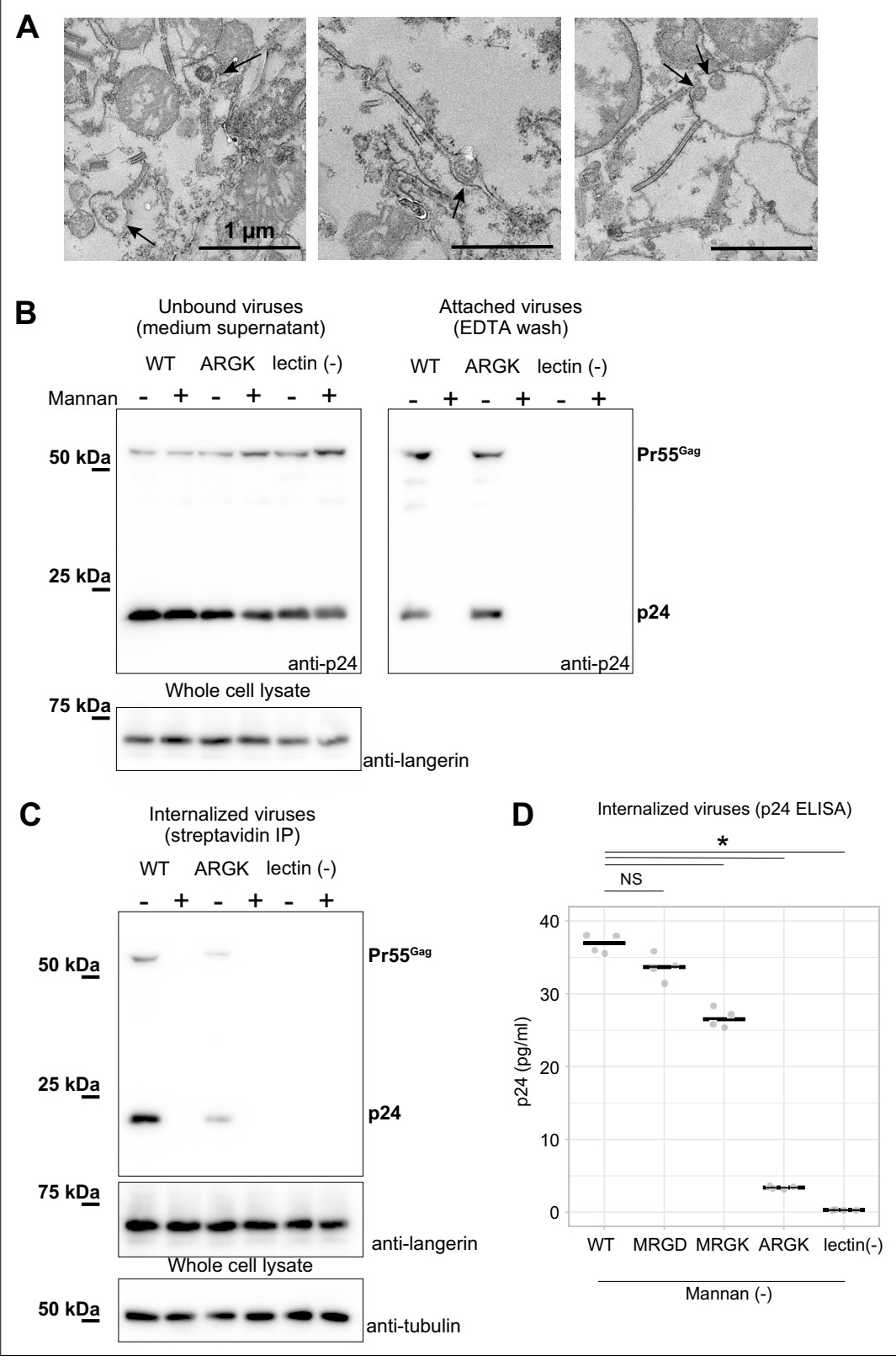

**Figure 5.** Mutation to the loop 258–263 inhibited the internalization of pseudoviruses. HIV-1 pseudoviruses were added to langerin-expressing 293T cells. Yeast mannan (10 µg/ml) was added to block the lectin-dependent binding of pseudoviruses. A langerin mutant lacking calcium binding ability (lectin (-)) was used as the negative control. (**A**) Immunoblots of pseudoviruses attached to the cell surface. Unbound and attached viruses were

*Figure 5 continued on next page*

*Figure 5 continued*

collected from the supernatant of the culture medium and TBS-EDTA buffer, respectively. Samples of unbound viruses were diluted 50-fold to adjust the band intensities. The expression levels of SNAP-tagged langerin show that the numbers of transfected cells were approximately the same in each experiment. Pr55$^{gag}$ and p24 indicate unprocessed and fully-processed capsid proteins, respectively. (**B**) Immunoblots of internalized pseudoviruses. Birbeck granules were isolated by precipitation using streptavidin-agarose, and intracellular viruses and langerin were detected by their respective antibodies. Tubulins in the whole-cell lysates were detected for loading controls. (**C**) Quantification of internalized viruses using p24 ELISA. Horizontal lines indicate the mean. NS and Asterisk indicate no significant difference and statistically significant differences (p=0.07 (MRGD); 9.4×10$^{-5}$ (MRGK); 9.9×10$^{-9}$ (ARGK); and 5.5×10$^{-9}$ (lectin(-))) calculated using Bonferroni-corrected Student's *t*-tests (N=4), respectively.

The online version of this article includes the following source data and figure supplement(s) for figure 5:

**Source data 1.** Original blot image of *Figure 5B* (right, anti-p24).

**Source data 2.** Annotated blot image of *Figure 5B* (right, anti-p24).

**Source data 3.** Original blot image of *Figure 5B* (right, anti-langerin).

**Source data 4.** Annotated blot image of *Figure 5B* (right, anti-langerin).

**Source data 5.** Original blot image of *Figure 5B* (left, anti-p24) and *Figure 5C* (anti-p24).

**Source data 6.** Annotated blot image of *Figure 5B* (left, anti-p24) and *Figure 5C* (anti-p24).

**Source data 7.** Original blot image of *Figure 5C* (anti-langerin).

**Source data 8.** Original blot image of *Figure 5C* (anti-tubulin).

**Source data 9.** Annotated blot images of *Figure 5C* (anti-langerin and anti-tubulin).

**Figure supplement 1.** Virus internalization by langerin.

**Figure supplement 1—source data 1.** Original blot image of *Figure 5—figure supplement 1A*.

**Figure supplement 1—source data 2.** Annotated blot image of *Figure 5—figure supplement 1A*.

**Figure supplement 1—source data 3.** Original blot image of *Figure 5—figure supplement 1C* (anti-langerin).

**Figure supplement 1—source data 4.** Original blot image of *Figure 5—figure supplement 1C* (anti-tubulin).

---

The coordinates and the rotation angles of the aligned subtomograms were exported to Relion-4.0. Initial pseudosubtomograms were generated with the box size of 128 pixels, the cropped box size of 64 pixels, and the binning factor of 4. The subtomograms were 3D refined using a spherical mask of 320 Å diameter. After 3D classification, 2×binned pseudosubtomograms were generated with the box size of 256 pixels and the cropped box size of 128 pixels based on the refined coordinates and angles and were 3D refined using the same spherical mask. Finally, un-binned pseudosubtomograms were generated with a box size of 256 pixels and a cropped box size of 128 pixels. 3D refinement was done using a tighter mask enclosing the central upright langerin molecule and three bound CRDs, and low-resolution subtomograms were removed by 3D classification. The alignment was further refined using CTF refinement and frame alignment. We initially refined the surrounding inverted langerin molecule by 3D refinement masking out the central upright molecule, but the refinement was biased toward the center. To improve the map quality of the inverted molecule, we applied 3D multi-body refinement (*Nakane et al., 2018*), which successfully compensated for the flexibility between the upright and the inverted molecules. For model building, the crystal structure of langerin trimer (*Feinberg et al., 2010*) (PDB ID: 3KQG) was fitted into the map using UCSF Chimera (*Pettersen et al., 2004*). Then, the fitted model was further refined using PHENIX real-space refinement tool with the secondary structure restraints and Ramachandran restraints on, and the refined model was validated using the comprehensive validation tool (*Adams et al., 2010*). The Coulombic surfaces rendering was conducted using UCSF Chimera and other model presentations, surface rendering, and morph movie generation were conducted using ChimeraX (*Goddard et al., 2018*). The data processing and the statistics of the model validation are summarized in *Figure 2—figure supplement 1A* and *Supplementary file 2*. Note that Alphafold (*Jumper et al., 2021*) failed to predict the docking between the loops 258–263 probably because the interaction requires carbohydrate ligands.

## Ultrathin section electron microscopy of Birbeck granules

Pellets of 293T cells with mannan-induced Birbeck granules were fixed with 4% paraformaldehyde for 1 hr at 4°C and stained with 1% osmium tetroxide and subsequently with 1% uranium acetate. After

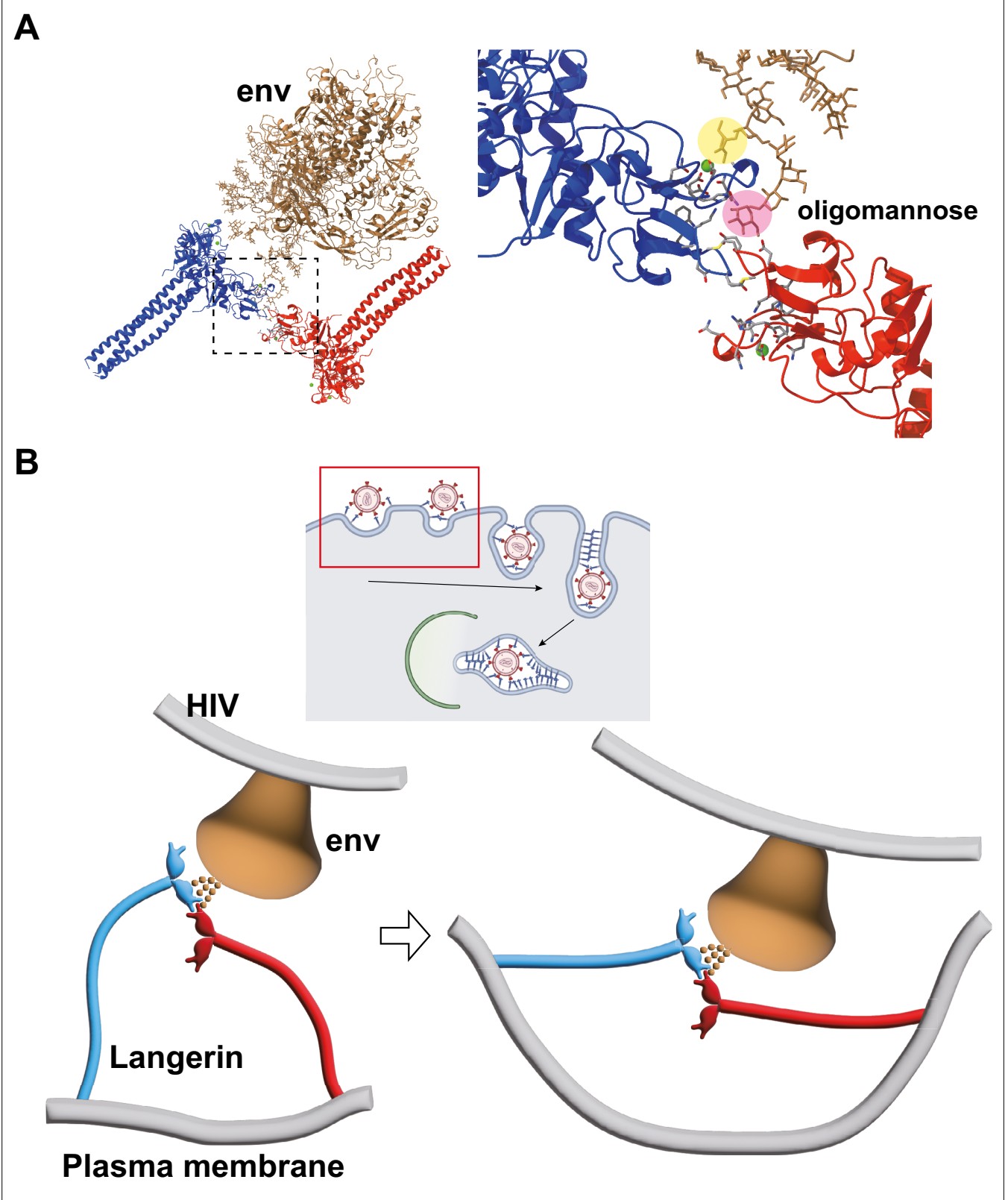

**Figure 6.** Model of HIV internalization by langerin. (**A**) Glycosylated structure of HIV-1 envelope (env, gray) (Constructed based on PDB ID: 5FYJ and 6ULC) bound to the langerin trimer complex. Branches of the high-mannose oligosaccharide interacting with the primary (yellow) and secondary (pink) carbohydrate-binding sites. The broken square indicates the position of the close-up view on the right. (**B**) Schematics of HIV internalization by langerin. The inset on the top shows an overview of antigen processing within a Langerhans cell. Viruses are captured by langerin, internalized into a Birbeck

*Figure 6 continued on next page*

*Figure 6 continued*

granule, and degraded by the autophagic system. The red square indicates the viral entry process described in the lower panels. The high-mannose oligosaccharide on HIV-1 env first binds to the primary carbohydrate-binding site, and one of the branches of the oligo-mannose weakly interacts with the secondary binding site. The flexible neck region allows docking of the loop 258–263 of an adjacent trimer into the secondary carbohydrate-binding cleft, reinforcing the binding of the oligo-mannose. As the elastic neck regions straighten, the plasma membrane is deformed into a lamellar invagination, which internalizes the virus.

dehydration in ethanol and acetone, the samples were embedded in Quetol 812 resin (Nissin EM, Tokyo, Japan). Sixty-nm-ultrathin sections were cut using a ULTRACUT microtome (Reichert Leica) and mounted onto Formvar-coated copper grid. Images were recorded using a JEM-2100F microscope (JEOL, Tokyo, Japan) at University of Yamanashi operated at 200 keV equipped with an F216 CMOS camera (TVIPS GmbH, Gauting, Germany). Length of Birbeck granules were measured using Fiji software (*Schindelin et al., 2012*), and plots were generated using PlotsOfData web tool (*Postma and Goedhart, 2019*). For total length of Birbeck granules per 100 $\mu m^2$, cross sections of cells with nuclei of >5 $\mu m$ in diameter were selected, and the sum of the lengths of Birbeck granules within the cells were measured. The areas of the cross sections were also measured using Fiji software. 100 $\mu m^2$ corresponds to the approximate mean of the cross-sectional area of one cell. All the measurements were repeated three times for obtaining technical replicates.

## Lentivirus preparation

Lentiviruses pseudotyped with HIV-1 env were prepared by transfecting the Lenti-X 293T cells with pCMV-dR8.3 Δvpr (Addgene plasmid #8455), pLOX-CW-tdTomato, and pCEP4-AD8gp160 (Adgene plasmid #123260) (*Stewart et al., 2003*; *Heredia et al., 2019*). The vector pLOX-CW-tdTomato was generated by replacing the GFP sequence of pLOX-CWgfp (Addgene plasmid #12241) (*Salmon et al., 2000*) with the tdTomato sequence derived from pCDH-EF1-Luc2-P2A-tdTomato, a gift from Kazuhiro Oka (Addgene plasmid # 72486). The transfections were carried out using the PEI-method with the ratio at PEI: dR8.3: tdTomato: gp160=4:1:1:1. 16 hr post-transfection, the cells were washed three times with PBS and incubated in a fresh medium for another 32 hr. The virus-containing medium was collected and removed cell debris by a 1000×g centrifuge and a 0.45-μm filtration (Millipore). The obtained supernatants were immediately used for subsequent assays.

## Virus attachment and internalization assay

293T cells were transfected with wild-type langerin, the ARGK mutant, and, as a negative control, a lectin (-) mutant lacking calcium-binding ability. The lectin (-) langerin mutant was generated by introducing three point-mutations at the calcium-binding site: Glu285Lys; Asn287Lys; and Glu293Lys (*Chatwell et al., 2008*). We conducted surface labeling of langerin-expressing cells and confirmed that these mutations did not alter the surface expression of langerin (*Figure 5—figure supplement 1A*). Forty-eight hr post-transfection, the cells were pre-treated with or without 10 μg/ml yeast mannan for 5 min at 37 °C. The cells were then incubated in the pseudovirus-containing supernatants plus 10 μM $O^6$-benzylguanine-biotin and 10 μg/ml DEAE-dextran (Sigma Aldrich, St. Louis, MO) with or without 10 μg/ml yeast mannan for 30 min at 37 °C. After the incubation, supernatant containing unbound viruses were collected. The cells were then washed with TBS-Ca buffer (20 mM Tris-HCl pH 7.4, 150 mM NaCl, 5 mM $CaCl_2$) three times, and the viruses attached to langerin at the cell surface was retrieved by washing with TBS-EDTA buffer (20 mM Tris-HCl pH 7.4, 150 mM NaCl, 10 mM EDTA). The collected supernatant of the medium and the TBS-EDTA buffer were centrifuged at 1000×g for 5 min to remove cell debris, and the viruses were sedimented by ultra-centrifugation at 100,000×g for 3 hr. After the TBS-EDTA washing, non-specifically bound viruses were removed by trypsinization using 0.25% (w/v) trypsin and 1 mM EDTA for 5 min at 37 °C as previously described (*Burkard et al., 2014*). Detached cells were washed twice with DMEM + 10% FBS, and five times with TBS-Ca. Internalized viruses within Birbeck granules were isolated from the washed cells by streptavidin-agarose precipitation as described above. Viruses, langerin, and tubulin were detected by immunoblotting using HIV-1 gag p24 antibody (1:1000 dilution, MAB7360, mouse monoclonal, R&D systems, Minneapolis, MN), human CD207 antibody (1:1000 dilution, 11841–1-AP, rabbit polyclonal, Proteintech Group Inc, Rosemont, IL), and α-tubulin antibody (1:3,000 dilution, B-512, mouse monoclonal, Sigma-Aldrich), respectively. ELSA was carried out using HIV Type 1 p24 Antigen ELISA kit (ZeptoMatrix, Buffalo, NY). The

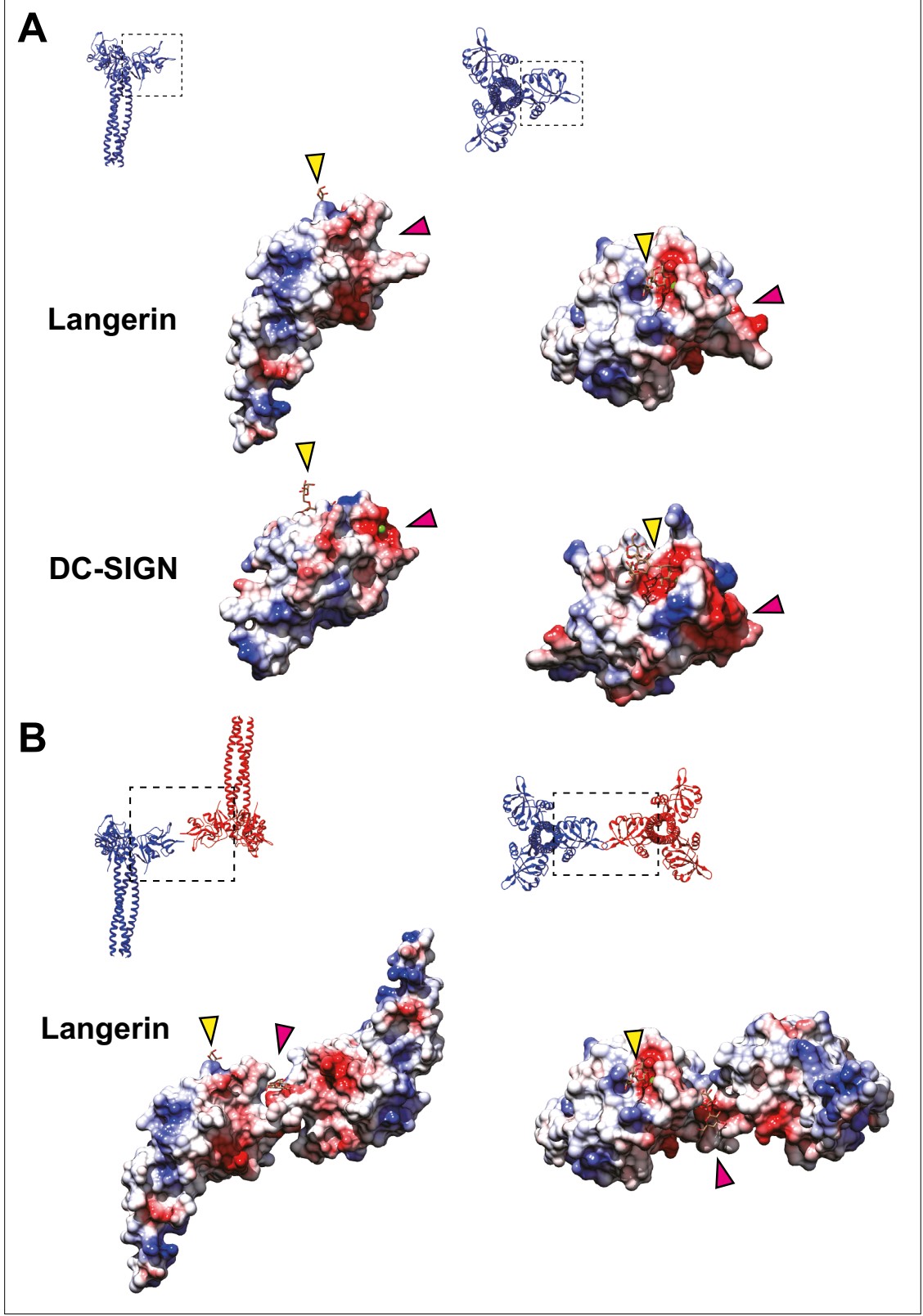

**Figure 7.** Coulombic surfaces of the carbohydrate-binding sites. (**A**) Comparison of the surface charge between the CRDs of langerin and DC-SIGN (PDB ID: 1SL4). Blue and red coloring indicates positive and negative charges, respectively. The primary and secondary carbohydrate-binding sites are indicated by yellow and pink arrowheads, respectively. The region of DC-SIGN corresponding to the secondary carbohydrate-binding sites of langerin is occupied by calcium ions (green) and does not form a cleft. (**B**) Surface model of the langerin trimer complex. Mannose ligands are placed based on

*Figure 7 continued on next page*

*Figure 7 continued*

a previously proposed model (*Chatwell et al., 2008*). (**A–B**) Broken squares indicate the position and the orientation of the side (left) and the top (right) views. Oligosaccharide ligands are shown in gray.

p24 antibody detected unprocessed (Pr55$^{gag}$) and fully processed (p24) capsid proteins (*Gonelli et al., 2019*). Band intensities were measured using ImageJ (*Schneider et al., 2012*). Surface labeling was carried out by treating the cells with 0.5 mM biotin-sulfo-osu (Dojindo, Osaka, Japan) for 30 min at 37°C. Cells were washed three times with TBS and collected by scraping. Surface-expressed proteins were extracted by incubating the cells in TBS plus 1% Triton X-100 for 1 hr at 4°C,and immunoprecipitated using streptavidin agarose. ELISA using p24 antibody showed that >99% of the viruses remained in the supernatant (*Figure 5—figure supplement 1B*). ELISA measurements were conducted twice to obtain biological replicates.

## Isolation of langerin-expressing stable cell lines

We isolated a cell line stably expressing langerin by selecting HEK293 cells using G418 (Sigma Aldrich). However, the expression levels of the isolated cell line was 10~20% of that of the transiently expressing cells (*Figure 5—figure supplement 1C*). These cells can form Birbeck granules in the presence of yeast mannan, but with lower efficiency (*Figure 5—figure supplement 1D, E*). Therefore, we chose to use the transient expression system in the main experiments.

## Acknowledgements

We thank Mrs Natsuko Maruyama (University of Yamanashi) for technical assistance. This research is partially supported by Platform Project for Supporting Drug Discovery and Life Science Research (Basis for Supporting Innovative Drug Discovery and Life Science Research (BINDS)) from Japan Agency for Medical Research and Development (AMED) under Grant Number JP19am0101115. This research was conducted using the Fujitsu PRIMERGY CX400M1/CX2550M5 (Oakbridge-CX) and the SGI Rackable C2112-4GP3/C1102-GP8 (Reedbush-H/L) in the Information Technology Center, The University of Tokyo. This work was supported by the Takeda Science Foundation (to T O), the Daiichi Sankyo Foundation of Life Science (to T O), the Japan Society for the Promotion of Science (KAKENHI Grant numbers JP21H02654 and JP22H05538 to T O), and the Naito Foundation (to T O). Molecular graphics and analyses performed with UCSF ChimeraX, developed by the Resource for Biocomputing, Visualization, and Informatics at the University of California, San Francisco, with support from National Institutes of Health R01-GM129325 and the Office of Cyber Infrastructure and Computational Biology, National Institute of Allergy and Infectious Diseases.

## Additional information

### Funding

| Funder | Grant reference number | Author |
| --- | --- | --- |
| Japan Society for the Promotion of Science | JP21H02654 | Toshiyuki Oda |
| Takeda Science Foundation | | Toshiyuki Oda |
| Daiichi Sankyo Foundation of Life Science | | Toshiyuki Oda |
| Naito Foundation | | Toshiyuki Oda |
| Japan Society for the Promotion of Science | JP22H05538 | Toshiyuki Oda |

The funders had no role in study design, data collection and interpretation, or the decision to submit the work for publication.

## Author contributions
Toshiyuki Oda, Conceptualization, Data curation, Formal analysis, Funding acquisition, Investigation, Methodology, Project administration, Writing - original draft; Haruaki Yanagisawa, Investigation, Methodology, Software, Validation; Hideyuki Shinmori, Investigation, Methodology; Youichi Ogawa, Tatsuyoshi Kawamura, Methodology, Supervision, Writing - review and editing

## Author ORCIDs
Toshiyuki Oda ⓘD http://orcid.org/0000-0001-8090-2159
Haruaki Yanagisawa ⓘD http://orcid.org/0000-0003-0313-2343
Hideyuki Shinmori ⓘD http://orcid.org/0000-0001-9864-8371
Youichi Ogawa ⓘD http://orcid.org/0000-0003-2635-888X

## Decision letter and Author response
Decision letter https://doi.org/10.7554/eLife.79990.sa1
Author response https://doi.org/10.7554/eLife.79990.sa2

## Additional files

### Supplementary files
• Supplementary file 1. Source code for generating the center coordinates within a polygon.
• Supplementary file 2. Table summary of data collection and model validation.
• MDAR checklist

### Data availability
The map and the model have been deposited in EMDB under the accession numbers: EMD-32906, and PDB 7WZ8.

The following datasets were generated:

| Author(s) | Year | Dataset title | Dataset URL | Database and Identifier |
|---|---|---|---|---|
| Oda T, Yanagisawa H | 2022 | Structure of human langerin complex in Birbeck granules | https://doi.org/10.2210/pdb7WZ8/pdb | Worldwide Protein Data Bank, 10.2210/pdb7WZ8/pdb |
| Oda T, Yanagisawa H | 2022 | Structure of human langerin complex in Birbeck granules | https://www.ebi.ac.uk/emdb/EMD-32906 | EMDB, EMD-32906 |

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
