## [Editor Report]

Langerhans cells, found in the epidermis and mucosal epithelium, capture pathogens via a C-type lectin called langerin. The capture leads to the formation of a membrane vesicle called a Birbeck granule. Here the authors use cryo-electron tomography to show, for the first time, how langerin forms a zippered lattice. They use their structure to design mutations to disrupt lattice formation and show that this interferes with HIV internalization.

---

## [Decision Letter]

[Editors' note: this paper was reviewed by Review Commons.]

---

## [Author Response]

We are glad that both of the reviewers appreciate the importance of our research. We have revised the text accordingly and already conducted all the requested experiments.

Reviewer #1Figure 1. The authors state that addition of mannan increases length of Birbeck granules however, no data are presented. It would make this more convincing when the length is compared between conditions with and without mannan (as shown in Figure 4, where the condition without mannan is lacking).

Thank you for pointing out the missing data. We added an EM image of Birbeck granules and quantification of Birbeck granules formation in the absence of mannan (Figure 4A-D).

Supp, Figure 1B perhaps as a panel in main figure as this is an important control to show that Birbeck granules are isolated.

We moved the supplemental figure 1B to main figure 1D.

Figure 4. Only the(total) length of Birbeck granules is taken into account, but not the number of Birbeck granules. Is it possible to quantify the number of Birbeck granules.

We added Figure 4D to show the number of Birbeck granules. Note that the difference in the number of Birbeck granules was less significant than that of total length because there were numerous short fragments in the mutant specimen.

Figure 5. Only the condition (ARGK) where there is virtually no Birbeck granules formation is included, however, is virus still internalized in the other conditions (MRGD or MRGK) as Birbeck granule formation was less effective but still present? It would be interesting to include those mutants. A more specific quantification would be by p24 ELISA. Is there a reason why immunoblotting has been chosen?In the supernatant condition, explain why the virus p24 seems less in the control condition whereas one would expect max concentration in that condition.

Thank you for suggesting the use of ELISA. We chose immunoblotting because of its higher sensitivity and lower cost. But ELISA is advantageous when it comes to comparing large number of samples. We performed p24 ELISA and quantified the virus internalization in all the mutants available (Figure 5D). As you pointed out, the transfer efficiency of the immunoblot in Figure 5A was not uniform across the membrane; Pr55 bands became denser toward the right, while p24 bands had a gradient in the opposite direction. The immunoblots and ELISA showed that about ~1% of the viruses were attached or internalized and ~99% did not interact with the cells. Thus, the attached/internalized viruses did not affect the amount of viruses in the supernatant. Results of ELISA also showed the amount of viruses in the supernatant were nearly equal among the samples (Figure 5—figure supplement 1B).

AbstractFirst sentence: not mucosal tissue but mucosal epithelium Last sentence: Virual should be viral

We corrected the typo. Thank you.

DiscussionThe last section comparing DC-SIGN and langerin is not clear and some overstatements are made. "Considering that DC-SIGN serves as an attachment receptor for viruses but not as an entry receptor, the possible structural coupling of lateral ligand binding and internalization implies that langerin functions as a more efficient entry receptor for viruses than DC-SIGN or other C-type lectins."It is not correct that langerin but not DC-SIGN can function as an entry receptor. DC-SIGN has been shown to facilitate infection of different viruses such DENV and ZIKV. In contrast, langerin can restrict viruses such as HIV-1 but also facilitate infection for example Influenza A and DENV. So attachment or entry is more likely a consequence of the internalization and dependence on pH changes for fusion as some viruses such as DENV fuse in acidic vesicles. This needs to be discussed more clearly.

Thank you for pointing out our wrong statement. We replaced the statement with weakened one as below:

Page 13, line 213: “The difference in the ligand-binding manner between langerin and DC-SIGN may contribute to their different carbohydrate recognition preferences (Valverde et al., 2020; Takahara et al., 2004).”

Reviewer #21) Langerin can exist on the cell surface and in Birbeck granules. They should examine langerin cell surface expression in the 3 states, wildtype, mutated and lectin –. Do the mutations change cell surface expression?

We performed surface labeling experiments and showed that those mutations did not affect surface expression of langerin (Figure 5—figure supplement 1A).

2) Birbeck granules are present in the absence of mannan and pathogens (see Pena-Cruz JCI 2018, PMID: 29723162). Thus, this suggests that Birbeck granules are present even without langerin clathrin coated pit internalization from the cell surface. How does their model account for this observation?

We think there are two possibilities:

1. Birbeck granules were shown to stem from endoplasmic reticulum (Valladeau et al. Immunity 2000; Lenormand et al. PlosONE 2013). Since the rER is the site of glycosylation, langerin is likely to capture the oligo-mannose-glycosylated proteins within the rER and form Birbeck granules.

2. Blood plasma proteins such as immunoglobulin D, immunoglobulin E, and apolipoprotein B-100 are reported to carry high-mannose glycans (Clerc et al. Glycoconj J. 2016). Those glycoproteins in the cell culture media can induce Birbeck granule formation.

3) Different cell types can have varied Langerin levels (see Pena-Cruz JCI 2018, PMID: 29723162). Is Birbeck granule formation depend on certain level of langerin expression? Do Birbeck granules form when Langerin is present at low as compared to high levels?

In the course of the experiments, we isolated a cell line stably expressing langerin. However, langerin expressing cells were extremely slow in proliferation and the expression levels were low. To answer this question, we recovered this “failed” stable cell line and found that the low langerin expressing cells can form Birbeck granules, but with lower efficiency (Figure 5—figure supplement 1C-E).

4) Authors use immunoblots to show that HIV is present in intra-cellular Langerin structures. It would be ideal to visualize HIV with presumably internal Birbeck granules using imaging techniques such as cryo-electron micrography or another form of high resolution imaging.

We conducted ultra-thin section electron microscopy of HIV-infected langerin-expressing cells and visualized HIV pseudo viruses within Birbeck granules (Figure 5A). Visualization of HIV containing Birbeck granules using cryo-electron microscopy is highly challenging because the current precision of cryo-FIB-SEM milling technique is too low to target a specific intracellular structure. We believe the EM images of Figure 5A provide sufficiently convincing evidences that HIV is internalized into Birbeck granules.